# Fossils from Mille-Logya, Afar, Ethiopia, elucidate the link between Pliocene environmental changes and *Homo* origins

Zeresenay Alemseged[1] ✉, Jonathan G. Wynn[2], Denis Geraads[3], Denne Reed[4], W. Andrew Barr[5], René Bobe [6,7,8], Shannon P. McPherron[9], Alan Deino[10], Mulugeta Alene[11], Mark J. Sier [12,13], Diana Roman[14] & Joseph Mohan[15]

Several hypotheses posit a link between the origin of *Homo* and climatic and environmental shifts between 3 and 2.5 Ma. Here we report on new results that shed light on the interplay between tectonics, basin migration and faunal change on the one hand and the fate of *Australopithecus afarensis* and the evolution of *Homo* on the other. Fieldwork at the new Mille-Logya site in the Afar, Ethiopia, dated to between 2.914 and 2.443 Ma, provides geological evidence for the northeast migration of the Hadar Basin, extending the record of this lacustrine basin to Mille-Logya. We have identified three new fossiliferous units, suggesting in situ faunal change within this interval. While the fauna in the older unit is comparable to that at Hadar and Dikika, the younger units contain species that indicate more open conditions along with remains of *Homo*. This suggests that *Homo* either emerged from *Australopithecus* during this interval or dispersed into the region as part of a fauna adapted to more open habitats.

[1] Department of Organismal Biology and Anatomy, University of Chicago, Chicago, IL 60637, USA. [2] Division of Earth Sciences, National Science Foundation, Alexandria, VA, USA. [3] CR2P, Sorbonne Universités, MNHN, CNRS, UPMC, CP 38, 8 rue Buffon, 75231, PARIS Cedex 05, France. [4] Department of Anthropology, University of Texas at Austin, Austin, TX 78712, USA. [5] Center for the Advanced Study of Human Paleobiology. Department of Anthropology, The George Washington University, Washington, DC 20052, USA. [6] Primate Models for Behavioural Evolution Lab, Institute of Cognitive & Evolutionary Anthropology, School of Anthropology, University of Oxford, Oxford, UK. [7] Gorongosa National Park, Sofala, Mozambique. [8] Interdisciplinary Center for Archaeology and Evolution of Human Behavior (ICArEHB), Universidade do Algarve, Faro, Portugal. [9] Department of Human Evolution, Max Planck Institute for Evolutionary Anthropology, Leipzig, Germany. [10] Berkeley Geochronology Center, 2455 Ridge Road, Berkeley, CA, USA. [11] School of Earth Sciences, Addis Ababa University, P. O. Box 1176Addis Ababa, Ethiopia. [12] CENIEH, Paseo Sierra de Atapuerca 3, 09002 Burgos, Spain. [13] Department of Earth Sciences, University of Oxford, South Parks Road, Oxford OX1 3AN, UK. [14] Earth and Planets Laboratory, Carnegie Institution for Science, Washington, DC 20015-1305, USA. [15] Climate Change Institute, University of Maine, Orono, ME 04469-5790, USA. ✉email: alemseged@uchicago.edu

For many decades, a disparity between the resolution of long and continuous marine paleoclimate records versus fragmentary and time-averaged terrestrial records has hampered our ability to establish precise links between human evolution and major environmental changes. However, a recent proliferation of fieldwork[1–4], new drilling campaigns targeting highly detailed and relatively continuous paleolake records[5,6], and novel geochemical approaches[7–10] are helping to assess terrestrial environmental dynamics at finer resolutions. Despite this progress, it remains the case that fauna, particularly hominins, are poorly sampled from the crucial time range between 3 and 2.5 Ma because fossiliferous sediments dating to this interval are rare. Although prolific deposits of the Omo-Turkana Basin in Ethiopia and Kenya do contain sediments from this interval, the hominin fossils are fragmentary and their taxonomic identities are uncertain. Sedimentary basins of the Afar in Ethiopia are highly fossiliferous, containing the most complete hominin record of the past 6 million years, alongside diverse faunas and well-established chronologies, but the 3–2.5 Ma interval is very poorly represented[11].

The Mille-Logya Project (MLP) is a new paleoanthropological site, dated from ca. 2.9 to 2.4 Ma, at the northeast end of the well-known Plio-Pleistocene sites in the Awash Valley of the Afar Regional State, Ethiopia (Fig. 1). Research at Mille-Logya started in 2012 and our team conducted systematic geological and paleontological surveys in 2014, 2015, 2016, 2018, and 2019. Here, we provide the first report on the geological and paleontological content of this site.

The fossiliferous sediments exposed at Mille-Logya are generally younger than most of other known research areas in the region. Thus, they offer a unique opportunity to elucidate major events in human evolution including the transition from *Australopithecus* to *Homo*, the emergence of *Paranthropus*, and the advent of manufactured stone tools[12,13]. Furthermore, although *Australopithecus afarensis* is the most abundantly preserved hominin from the region between 3.8 and 2.9 Ma, its fate is largely unknown because of a regional hiatus in the sedimentary record of the Hadar Basin between 2.9 and 2.7 Ma[14]. Above this unconformity, the sediments of the Busidima Formation fill a local half graben near the western escarpment of the Ethiopian Rift[15]. Compared to the largely lacustrine and peri-lacustrine Hadar Formation, the Busidima Formation exhibits a very different style of deposition, characterized by low sedimentation rates in almost exclusively episodic, high-energy fluvial settings, resulting in relatively poor fossil preservation. Meanwhile, during this same interval (2.9–2.7 Ma), lacustrine and peri-lacustrine sediments were deposited at Mille-Logya and continue into younger horizons.

Our geological work offers new evidence for the northeast migration of the Hadar Basin, expanding our knowledge of the history of the basin substantially. Three new fossiliferous horizons with differing faunal composition have been identified suggesting in situ faunal change. While the fauna in the older unit is comparable to that at Hadar and Dikika, the younger units contain species that indicate more open conditions along with remains of *Homo*. New data from Mille-Logya reveal how hominins and other fauna responded to environmental changes during this key period. Our results show a connection between geotectonics, sedimentary basin migration and an in situ faunal change. We also provide new evidence that could potentially explain what happened to *Australopithecus afarensis* after 2.9 Ma and what caused the dispersal to or the emergence of *Homo* in the region.

## Results

**Geology**. Stratigraphic surveys of the Mille-Logya area were geared toward broadening our understanding of the geological history of the region. To achieve this, it is crucial not only to establish chronological relationships between fossiliferous sites, but to investigate stratigraphic, structural and facies relationships between discontinuous exposures of sedimentary basins[16,17]. Early geological maps of the region showed isolated Plio-Pleistocene sediments within the new site, amidst basalt flows attributed to the Afar Stratoid Series[18,19]. In these broad-scale maps, the sediments were attributed to undifferentiated Quaternary strata or the White Series (Enkafala Beds; both mapping units, were also used to indicate outcrops of the Hadar Formation[20–22], the latter having been much more thoroughly scrutinized since initial fossil discoveries at the Hadar site[23–30]). Sediments in areas nearby[11,31,32] broadly bracket and partly overlap with the strata of the Hadar Formation. Our work has identified three new fossiliferous stratigraphic units expanding our knowledge of the geological history of the region and providing context for our faunal and basin analysis.

**Stratigraphy**. Sedimentary exposures in the Mille-Logya area provide access to generally disconnected sections of up to ~60 m in total thickness. Between these discontinuous exposures, extensive colluvial cover of volcanic, boulder- to cobble-sized material obscures most outcrop. Furthermore, a number of post-depositional faults divide the exposures into disconnected fault-bounded blocks. Hence, our stratigraphic interpretations of relationships between sections are presently based on widespread marker beds, chemical groupings of interfingered basalts and tephras, nine new $^{40}$Ar/$^{39}$Ar dates, and several magnetostratigraphic sections. These observations are sufficient to describe the overall stratigraphy, and to divide the sedimentary strata into three main fossiliferous intervals each exposed at one of the three main areas: Gafura, Seraitu, and Uraitele (Fig. 2). In this report, we designate these as informal stratigraphic units, with the aim of formalizing a regional lithostratigraphic terminology in future work, building on these presently informal terminologies.

The lowest stratigraphic unit, Gafura, (Fig. 2) begins with a sequence of thick, columnar-jointed basalt flows with intra-flow residual paleosols developed on the basalts. The Gafura sediments are poorly exposed, but occur along the southwestern flank of Iki-Ilu Ridge (Fig. 1), and are best represented by a section exposed at Sidiha Koma (section JGW15-1). The upper surface of the basalt flow at the base of this zone forms a broad low-lying surface, dissected by the Gafura River, extending into the base of the Daamé Valley. This sequence of basalts defines the GFB-I and GFB-II groups (GFB = Gafura Basalts); it underlies the main sedimentary sequence and is thus stratigraphically distinct from the overlying flows represented as the Afar Stratoid Series. Within the lowermost exposures of the Gafura Basalts, a normal to reverse magnetostratigraphic reversal is recorded (see Supplementary Table 5). Given the age constraints of overlying strata, this reversal must be equal to or older than the age of the base of the Kaena Chron (3.127 Ma)[33].

The transition to overlying sediments of the Sidiha Koma area is marked by mudstones with ferruginized burrows interspersed with thin, poorly-sorted sands with a framework of basaltic lithic grains, occasionally containing abundant gastropods, and some bivalves. Near the top of Gafura sediments, additional basalt flows overlie the sediments locally, although these have not yet been attributed to one of the geochemically-defined groups (see Supplementary Figs. 1, 2 and Supplementary Table 1). The fossiliferous sediments of Gafura underlie a widespread diatomaceous unit, the Iki-Ilu Diatomite, which can be mapped in regionally extensive exposures along the southwestern flank of Iki-Ilu Ridge, across its southern tip, and into the floor of the Seraitu Valley, making this a practical stratigraphic boundary.

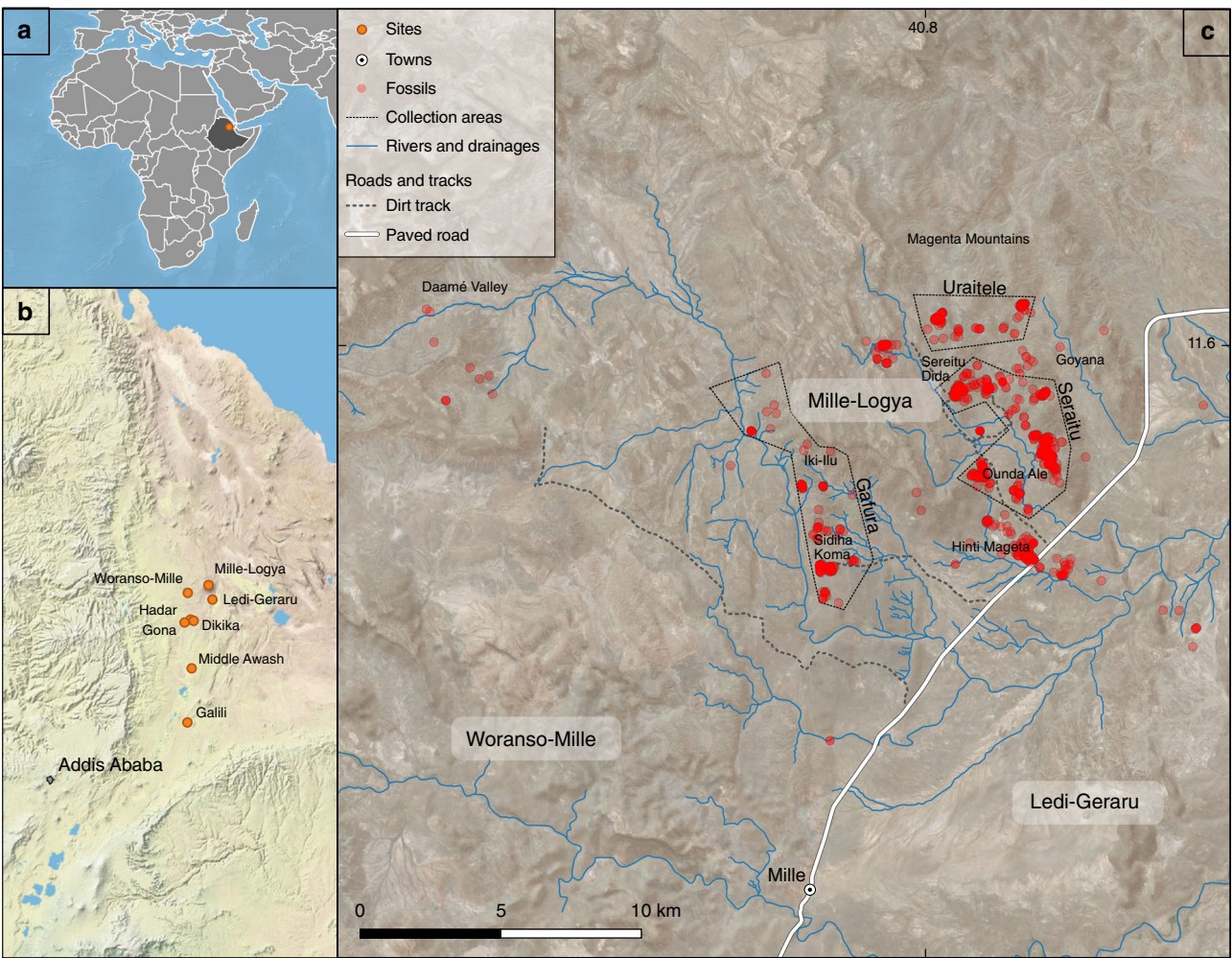

**Fig. 1 Location of the Mille-Logya Project (MLP) area within the greater Afar sedimentary basin, Afar, Ethiopia.** Panel **a** shows the location of MLP within Ethiopia; **b** the localtion of MLP relative to other major Plio-Pleistocene hominin sites in the Afar triangle; **c** the distribution of individual fossils and the three main collection zones (Gafura, Seraitu, and Uraitele) The maps in this figure were generated with the standard distribution of QGIS version 3.10.2.

The Hinti Mageta Tuff ($2.914 \pm 0.036$ Ma; see Supplementary Figs. 3–5 and Supplementary Tables 4, 5), preserved within the Iki-Ilu Diatomite, provides an upper age limit to the Gafura sediments.

The middle stratigraphic unit is represented by the Seraitu lake beds, which often form bare, steep cliffs of largely mudstone outcrops representing lacustrine deposition. We use the base of the Iki-Ilu Diatomite as the lower boundary of this zone, although most sections cannot be mapped in continuity within a measured stratigraphic distance to the diatomite. The upper boundary of this zone may not be defined by a single widespread marker but can be locally taken as the stratigraphically lowest basalt flow with chemical composition characteristic of the UGB Group (Uraitele-Garsele Dora Group), which may consist of several different flow units. The UGB is frequently accompanied by an overlying, distinctive and widespread air-fall tuff with well-preserved glass and small lapilli-sized pumices, the Goyana Tuff (see Supplementary Table 2). Thus, the presence of one or both of these markers provides a working stratigraphic definition.

The sediments of the Seraitu lake beds are predominantly laminated clays which often contain abundant ostracods, gastropods and some bivalves, as well as plant fragments and fish remains. Diatoms in the Iki-Ilu Diatomite are somewhat recrystallized but identifiable to the genus *Aulacoseira*[34,35].

Tephras are also numerous in the lake beds but characteristically thin (<30 cm), often air-fall occurrences, in which the primary glass is altered. Despite this, abundant feldspar crystal populations are preserved, providing two of the $^{40}Ar/^{39}Ar$ dates reported here. Besides the Hinti Mageta Tuff at the base of the Seraitu lake beds, two tuffs within the lake sediments provide precise ages: MLP14/SR-6 at $2.576 \pm 0.008$ Ma and MLP14/GOY-2 at $2.485 \pm 0.018$ Ma. In addition to the chronological information from these markers, two sections within the Seraitu lake beds zone record a magnetostratigraphic reversal which we interpret to be the Gauss/Matuyama, dated to 2.59 Ma (Fig. 1; section JGW14-10 and in section JGW14-6; see Supplementary Figs. 6–8, Supplementary Table 5).

The third unit, Uraitele, includes limited sedimentary exposures in-between extensive and thick basalt flows of the UGB, GYB-I, and GYB-II groups, which outcrop in the Goyana and Uraitele areas. The sediments include some lenticular sandstones interpreted as fluvial channels, but are predominantly laminated mudstones with occasional gastropod and bivalve bearing sandstones formed on surfaces of the UGB basalt or within the mudstones. The upper boundary of the Uraitele zone is as yet undefined, as the section continues in a thick sequence of numerous basalt flows that extend into the ridges of the Magenta Mountains at the northern extent of the area (mapped as the Afar

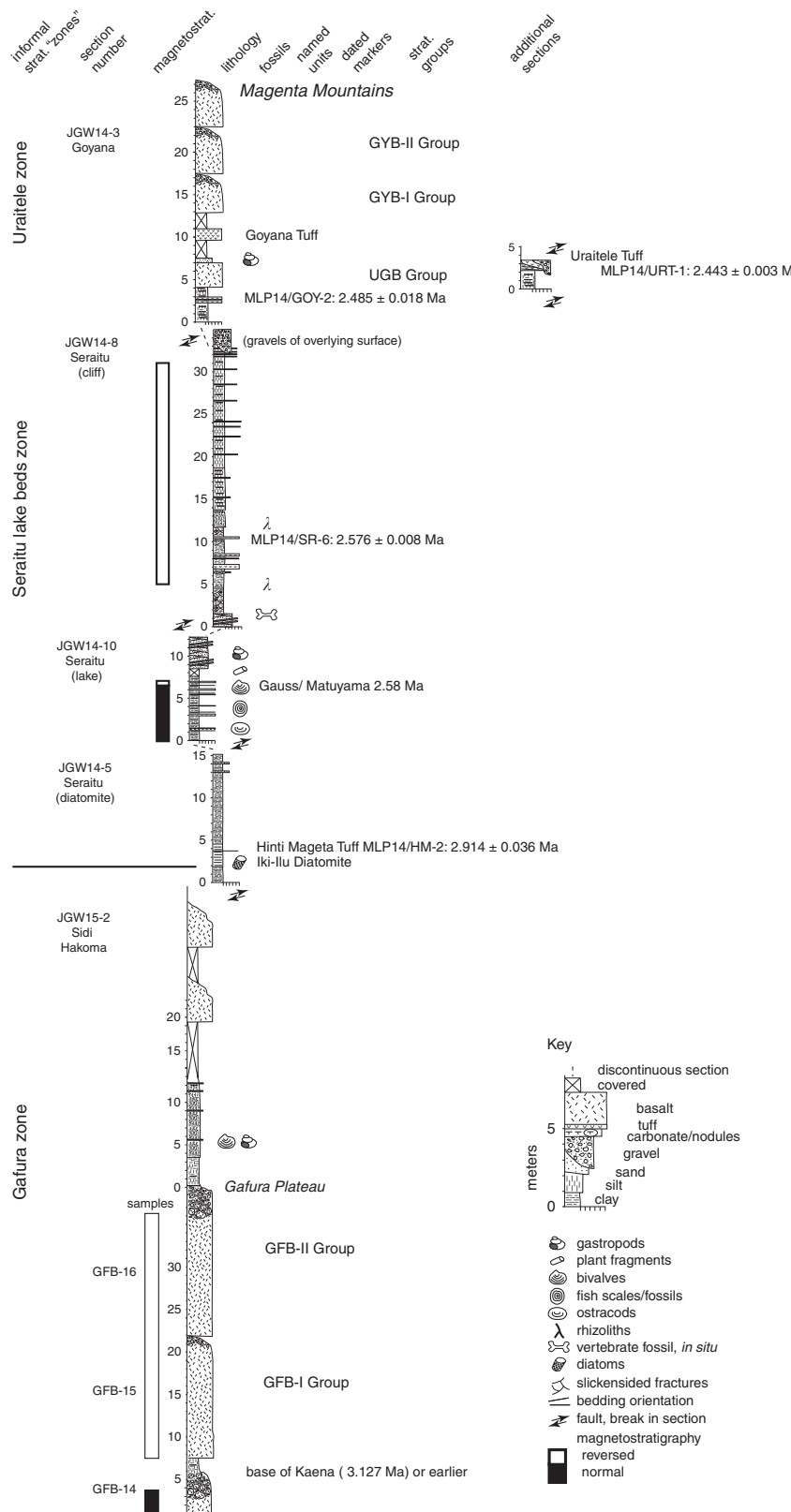

**Fig. 2** Stratigraphic sections, sedimentary content volcanic markers, and relationships among the three fossiliferous zones (Gafura, Seraitu, and Uraitele) at the Mille-Logya Research area.

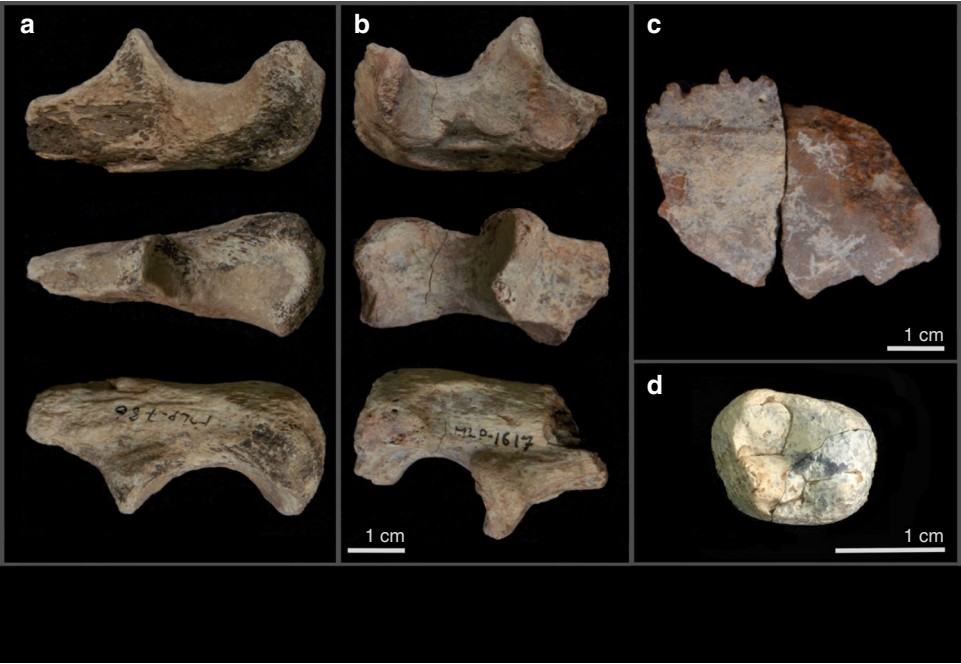

**Fig. 3 Hominin remains from the MLP area. a**, **b** are left and right proximal ulnae (MLP-786 & MLP-1617, respectively from two different localities, thus not from the same individual); **c** = calvarium fragment (MLP-1469); **d** = a diagnostic and complete upper second molar crown (MLP-1549).

Stratoid Series[20,21]). The uppermost flows form two chemically-distinct groups, termed the GYB-I and GYB-II groups (Supplementary Table 1).

As with the Gafura area, the sediments of the Uraitele area contain a number of reworked vitric tuffs generally lacking large feldspar populations, but having distinctive chemical composition with no known correlates from the Awash Valley (see Supplementary Table 2). One of these vitric tephras, the Uraitele Tuff, has also produced populations of feldspars suitable for dating, and represents the most precise age of those presently analyzed from the Mille-Logya area: 2.443 ± 0.003 Ma.

**Depositional history.** Given the above stratigraphic sequence, we can make some important interpretations of the basin's history. Prior to ~3 Ma, there is no evidence of an active depositional basin in the Mille-Logya region. Thus, during the period characteristic of most active lacustrine sedimentation at Hadar and Dikika (~3.6–3 Ma)[14,15], the sequence of Gafura Basalts and residual paleosols at Mille-Logya suggests subaerial emplacement of basaltic lavas followed by periods of non-deposition and pedogenesis. The local onset of active subsidence and sedimentation is marked by early onset of shoreline and shallow lacustrine deposits overlying the uppermost GFB-II Group. The subsequent lacustrine sequence culminates in a deep, well-mixed lake represented by an *Aulacoseira*-dominated diatom facies within the Iki-Ilu Diatomite and invertebrate fossil rich mudstones preserved throughout the Seraitu lake beds zone. Few terrestrial indicators are present except occasional coarser-grained facies suggestive of shorelines or brief subaerial exposure, where occasional rhizoliths are preserved and vertebrate fossils are slightly more common. This lacustrine setting is persistent throughout the exposures and continues into the overlying Uraitele sediments, the lowermost of which are characterized by gastropod-bearing shoreline facies developed on the surface of basalt flows. Ultimately, the lacustrine phase ends with thin intervening sediments between basalt flows of the GYB Group, which have been associated with the fissural system of the axial rift[36].

**Archaeology.** Archaeological survey was conducted in conjunction with paleontological reconnaissance. Archaeological visibility is extremely low due to the combination of a thick colluvial cover, relatively few exposed sections and the fact that most of the sedimentary deposits are lacustrine in origin. Nevertheless, all of the fossiliferous localities and their surroundings were examined on multiple occasions. In general, lithic artifacts are infrequent and scattered at very low density. The only exception comes from Seraitu Dida where slopes on two adjacent ridges with sediments above the Uraitele Tuff have numerous handaxes and Levallois cores and flakes made on a fairly consistent coarse-grained but good quality volcanic material. While artifact densities were relatively high in this area, no concentrations suggested a source for this material that has temporal constraint. Nevertheless, we excavated three trenches at the crest of one of the ridges above the Uraitele Tuff. In one of these trenches, about one meter into a layer of gravel, we found a single Levallois flake. A maximum age is provided by the Uraitele Tuff (2.443 Ma) but the minimum age remains unconstrained. Archeological exploration will continue.

**Paleontology.** The aforementioned stratigraphic setting provides a framework to interpret fossil data recovered from the three units. Fossil concentrations at Mille-Logya are sparse and relatively difficult to locate. Yet, after four field seasons, the fossil collection currently includes 2287 specimens, of which 1835 were collected while the rest were observed and documented on site (Table 1). Fossil collections at MLP followed a standardized protocol[37] in order to minimize collection bias. The identifiable specimens comprise 62 Cercopithecidae, 4 Hominidae, 33 Proboscidea, 10 Camelidae, 165 Suidae, 135 Hippopotamidae, 36 Giraffidae, 944 Bovidae, 218 Equidae, 21 Rhinocerotidae, 20 Carnivora, 17 birds, and some rodents, fishes, turtles, and crocodiles. Below is a summary description of the major faunal elements.

**Hominidae**: Hominins were recovered from four different localities and are represented by a left and right proximal ulnae (MLP-786 & MLP-1617 respectively (Fig. 3a, b): 2.6–2.8 Ma; from two different localities, thus not from the same individual), a

**Table 1 Distribution of vertebrate taxa across the MLP main groups of exposures.**

|  | Gafura | Seraitu | Uraitele |
|---|---|---|---|
| *Deinotherium bozasi* | – | X | X |
| *Elephas recki* | X | X | X |
| *Orycteropus* sp. | – | X | – |
| *Camelus grattardi* | – | X | – |
| Aff. *Hippopotamus protamphibius* | X | X | X |
| *Kolpochoerus limnetes* | X | X | X |
| *Kolpochoerus* n.sp.? | – | X | – |
| *Notochoerus euilus* | X | X | X |
| *Giraffa jumae/stillei* | X | X | X |
| *Giraffa pygmaea* | – | – | X |
| *Sivatherium maurusium* | X | X | X |
| *Aepyceros* sp. | X | X | X |
| *Connochaetes* sp. | – | – | X |
| *Damaliscus* cf. *ademassui* | – | – | X |
| *Damalborea* sp. | – | X | – |
| Alcelaphini indet., very small | X | X | X |
| *Gazella harmonae* | X | X | ? |
| Bovini indet. | X | X | X |
| *Pelorovis* cf. *kaisensis* | – | X | – |
| *Kobus sigmoidalis* | ? | X | X |
| *Kobus* cf. *oricornus* | X | X | – |
| *Tragelaphus* aff. *lockwoodi* | – | ? | X |
| *Tragelaphus nakuae* | ? | X | X |
| *Ceratotherium* sp. | X | X | – |
| *Diceros* sp. | – | X | – |
| Hipparionini sp. | X | X | X |
| *Crocuta eturono* | – | ? | X |
| cf. *Dinofelis* sp. | – | X | X |
| Felidae indet., serval size | – | X | – |
| Cercopithecidae indet. | X | X | X |
| *Theropithecus* sp. | X | X | X |
| *Homo* sp. | – | – | X |
| *Crocodylus* sp. | X | X | X |
| *Euthecodon* sp. | – | X | X |
| Chelonia | X | X | X |
| Anatidae indet. | – | – | X |
| *Struthio* sp. | X | – | X |
| Siluriformes | – | X | – |

calvarium fragment (MLP-1469 (Fig. 3c): 2.6–2.8 Ma) and a diagnostic and complete upper second molar crown (MLP-1549 (Fig. 3d); 2.4–2.5 Ma). The molar, found in two pieces that refit cleanly, measures 14 mm and 12.5 mm buccolingually and mesiodistally, respectively, falling within the known range of *A. afarensis* as well as early *Homo* as represented by A.L. 666-1 from the younger horizons at Hadar dated at 2.33 Ma. The buccolingual and mesiodistal dimensions overlap with those of early *Homo* and are closest to the mean values (Fig. 4 a, b; Supplementary Fig. 10). The occlusal outline, which is dominated by the two mesial cusps, is rhomboidal with the longest axis running from the distolingual to mesiobuccal corners. The distobuccal corner is truncated. The tooth is moderately worn with no cuspal dentine exposure. The lingual wear flattens the protocone and hypocone and polishes the lingual margin leading to a rather homogenized region of the lingual half of the tooth. In contrast, the paracone and metacone are not as worn and the buccal margin is still sharp. The distal marginal ridge and distal fovea are quite perceptible, but the mesial marginal ridge is largely worn down leaving only a hint of the mesial fovea. In occlusal view, running buccal to the protocone and distal to the paracone is a large buccal groove that dominates other grooves

and is positioned mesiobuccal to a lingual groove that is much smaller.

An asymmetric and rhomboidal occlusal outline of the upper second molar has been reported to characterize *Homo erectus* and *H. habilis*[38,39] but is rare in *A. afarensis*[36]. The MLP M2 possesses these features but is buccolingually broad unlike *Homo erectus*. Based on size and average enamel thickness (Fig. 4 a, b; Supplementary Fig. 11) in addition to diagnostic occlusal features, we attribute it to *Homo* sp. With an age of 2.4–2.5 Ma, this molar represents one of the oldest specimens of this genus and expands the earliest *Homo* sample from the Afar, which currently includes only LD 350-1 from Ledi Geraru at 2.8 Ma and A.L. 666-1 from Hadar at 2.33 Ma.

The calvarium fragment is probably from the parietal; the bone is relatively thin and different from what is seen in middle Pleistocene specimens such as Bodo. MLP-1617 and MLP-786 are fragmentary proximal ulnae mainly preserving the olecranon and the trochlear notch. The two specimens differ in size and degree of preservation. While MLP-786 is larger, MLP-1617 is better preserved, especially its trochlear notch where the maximum mediolateral breadth, including the radial notch, is ca. 28 mm. In MLP-786, most of the distal aspect of the trochlear notch including the radial notch is broken away. In both, various quadrants of the trochlear notch are mildly concave. Maximum radial notch dimensions are 13.4 mm anteroposteriorly and 9.4 mm superoinferiorly in MLP-1617. In both specimens, the olecranon process is moderately projected proximally and slightly more pronounced in MLP-786, as is seen in other hominins. The olecranon proximodistal height is 7.3 mm and 8.6 mm respectively and similar to that of A.L. 438-1 (*A. afarensis*). The trochlear keel is mild in MLP-1617 and not perceptible in MLP-786. Posteriorly, MLP-1617 and MLP-786 measure 15.3 mm and 18.5 mm, respectively at the middle of the trochlear notch and proximally they measure 20.8 mm and 23.6 mm. The two ulnae can readily be assigned to Hominini based on the extent of the olecranon process and the orientation and extent of the trochlear and radial notches. Many features and dimensions discussed by Drapeau[40] to characterize most fossil hominins are preserved (mainly in MLP-1617) but further discrimination is not possible and these bones are therefore attributed to Hominini indet.

**Cercopithecidae**: cercopithecines, especially *Theropithecus*, are abundant though mostly represented by fragmentary teeth and extremities of postcranials. The MLP *Theropithecus* is characterized by diagnostic high crowned, bilophodont molars with deeply incised notches and clefts. Based on limited existing material, molar size appears larger than in representatives of *T. oswaldi* from the older members of the Hadar Formation. Some teeth displaying similar features are smaller; we interpret them as females of the same species. There are also several teeth that do not display these diagnostic *Theropithecus* features, which we assign to Cercopithecinae indet. pending recovery of more complete specimens.

**Proboscidea**: enamel fragments belonging to *Deinotherium* are frequently encountered, and a nearly complete skull was excavated at Uraitele. Dental elements and postcranials of Elephantidae are common as well, and some were collected; a complete M3 from Uraitele best matches *Elephas recki shungurensis*[41] known from Omo Shungura Members C to F, but there are no clear boundaries between successive subspecies to allow more precise biochronological attribution.

**Tubulidentata**: a single mandibular fragment belongs to an aardvark (*Orycteropus*).

**Camelidae**: this family is extremely rare in the East African Plio-Pleistocene, and the discovery of ten specimens at Mille-Logya is remarkable. One of them is the only partial skull known

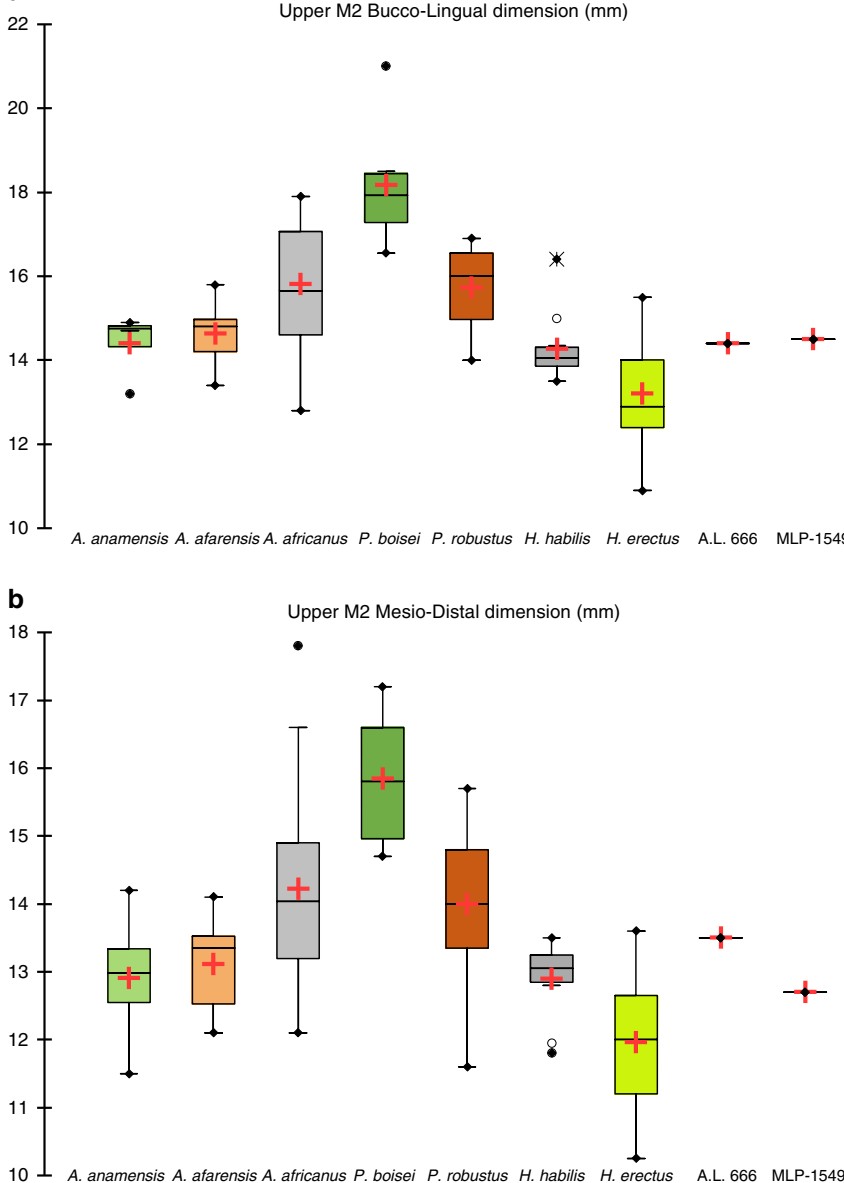

**Fig. 4 Molar crown dimensions.** Buccolingual and mesiodistal dimensions of MLP-1549 compared to values in *Au. anamensis*, *Au. afarensis*, *Au. africanus*, *P. boisei*, *P. robustus*, *H. habilis*, *H. erectus* & A.L. 666-1(*Homo sp.* from Hadar dated to 2.33 Ma). **a** upper second molar bucco-lingual dimensions in mm; **b** upper second molar mesio-distal dimensions in mm. Source data are provided as a Source Data file.

from this part of Africa to date and detailed analysis of this skull has been published elsewhere[42].

**Hippopotamidae**: though remains of this group are common throughout the sequence, there is only one species; it has a hexaprotodont dentition, with a slightly smaller i2 than i1 and i3, thus resembling early forms of the 'aff. *Hippopotamus protamphibius*'[43]. It is clearly different from *Hexaprotodon bruneti*[44], from the Hata Member of the Bouri Formation at ca. 2.5 Ma, which has a very large i3. Yet, formal identification must await the revision of the Turkana and Hadar material. No tetraprotodont dentition has been recovered from the MLP area.

**Suidae**: *Notochoerus* is the most common genus. Several complete third molars, mostly from the Gafura unit, are smaller than those of *N. scotti* from Omo C and later members, and match those of *N. euilus* from Omo B or the Hadar Formation. The morphology and mesiodistal length of their molars are similar to those in *N. clarki*, which coexists with *N. scotti* at Omo,

but are broader and we attribute them to *N. euilus*. *Kolpochoerus* is less common and all specimens are of comparable size. Dimensions of the third molars, mostly from the Seraitu lake beds, are close to the upper end of the range for *K. afarensis* from the Hadar Formation, or to the lower end of the range for *K. limnetes* from Shungura D-E, and are closest to specimens from Shungura B and C. They are also somewhat larger than those of *K. philippi*[45], from Matabaietu at ca. 2.5 Ma. There is no definite evidence of *Nyanzachoerus*, nor of *Metridiochoerus*.

**Giraffidae**: both *Giraffa* and *Sivatherium* are represented, but are rare. The relatively more common *Giraffa* is not particularly large, and a second, smaller species (*G.* cf. *gracilis*) is also present.

**Bovidae**: Bovids are by far the most common mammals, and almost half of the identifiable specimens, mostly represented by teeth, belong to Alcelaphini. They are followed in decreasing order of abundance by the Reduncini, Bovini, Aepycerotini, Antilopini, and Tragelaphini. Horn-cores are encountered

relatively frequently, but are seldom associated with other cranial parts. An alcelaphin that resembles *Damaliscus ademassui* from Gamedah dated to ca. 2.5 Ma[46] and perhaps a primitive wildebeest (*Connochaetes* sp.) are present at Uraitele. We assign the most common alcelaphin to *Damalborea*, a genus that is present throughout the Hadar Formation[47]. Although variation at Hadar is great, the MLP form is distinctive in its short, twisted horn-cores with homonymous torsion. It could be an evolved form of *D. grayi* from the Denen Dora Member at Hadar. An unidentified, very small alcelaphin is reminiscent of the one that first appears in the Kada Hadar 2 submember[47]. There are at least two species of reduncins. The less common one is probably *Kobus sigmoidalis*, best known from the Turkana basin and recently reported from Ledi-Geraru[11]. The specific identity of the more common reduncin is not clear; it resembles *K. oricornus* from Omo Shungura[48], West Turkana, Koobi Fora[49], and Hadar[47]. Surprisingly, this taxon is absent in the nearby Ledi-Geraru[11]. A Bovini horn-core from the middle part of the section is comparable to the type of *Pelorovis kaisensis*[49], from Kaiso village in Uganda, ca. 2.5 Ma. *Aepyceros* is common but species identification is difficult due to incompleteness. One of the horn-cores is larger than those from the Kada Hadar Member. *Gazella*, the only antilopin so far recovered, is represented by several long and slender horn-cores resembling *G. harmonae* from the Kada Hadar Member at Hadar, and probably Omo Shungura Member F and Olduvai Bed I. Most tragelaphin horn-cores are from the younger part of the exposures and resemble *Tragelaphus nakuae* in their moderate torsion and in the presence of a low supra-occipital ridge of braincase. A very long horn-core is reminiscent of a specimen from Omo-160 in Shungura Member C[48]. Overall, the tragelaphin material suggests an age of 2.6–2.3 Ma. A second and rare species from the middle part of the sequence is similar to *T. gaudryi* of which an ancestral form appears in Omo Shungura Member C.

**Rhinocerotidae**: Though rhinos are rare here, both the grazer *Ceratotherium* and the browser (or mixed-feeder) *Diceros* are encountered.

**Equidae**: Equids are fragmentary but quite common. We tentatively attribute all remains to a single species of hipparion. A complete set of upper incisors shows that the I3 is not reduced, and the lingual grooves are shallow, in contrast to what is seen in derived hipparions of the *cornelianus* group. A remarkable feature is the absence or poor development of the ectostylid on many lower teeth. A moderately worn and well-preserved set of molars shows no ectostylids at all. Postcranial dimensions are close to the lower end of the Hadar range[50,51] where skulls show that at least two species are represented[50,51].

**Carnivora**: Some postcranials potentially representing multiple taxa belong to Hyaenidae. Two metapodials and a tooth belong to a felid, cf. *Dinofelis*. In addition, a distal radius belongs to a serval-sized felid.

**Aves**: A. Louchart (pers. com.) identified a large ostrich and a member of the Anatidae, perhaps *Sarkidiornis melanotos* or *Plectropterus gambensis*. Large ostriches have been mentioned from a number of Pleistocene Old World sites; they are likely attributable to *Struthio asiaticus*.

**Crocodylidae**: Crocodile teeth are widespread, and a few specimens represent *Euthecodon*.

**Fishes**: K. Stewart (pers. com.) identified bagrid and clariid fishes.

**Biochronology and paleoenvironment**. The Hadar Formation fauna, documented in the nearby sites of Hadar, Dikika, and Ledi-Geraru, has been widely studied and offers a very good reference for the new material from Mille-Logya. The Mille-Logya fauna points to a generally younger, late Pliocene age but shares a number of taxa with those from the Hadar Formation, where many have a wide chronological range. Of biochronological significance are the antelopes, *Damalborea* and *Kobus* cf. *oricornus*. *Gazella harmonae* is also shared with Hadar, although this species has wide chronological and geographic ranges. Another indicator of a similar age is the hexaprotodont hippopotamid, present in the middle part of the sequence at Mille-Logya. The suids also fall largely within the size range seen in the Hadar Formation, but most diagnostic specimens are encountered in the lower and middle parts of the sequence. It should be noted, however, that *Kolpochoerus* from the Seraitu lake beds is more consistent with the older absolute ages of this unit than with the younger ones. The absence of *Nyanzachoerus* suggests that the MLP assemblage postdates most of the fauna from Hadar. Although the above is generally true, there are differences within the Mille-Logya fauna indicating that sites in the southern portion of the research area are older than those in the north.

As in most African Pliocene sites, bovids are the most common group followed by equids and suids. Primates are fairly common but rare compared to those at Hadar and Dikika. One striking feature of the Mille-Logya fauna is the high prevalence of equids, particularly relative to suids. At Hadar and Dikika the reverse is consistently the case. While this is true when the whole Mille-Logya assemblage is considered as a unit, looking at the different horizons reveals a different pattern. Gafura (~2.9 to 2.8 Ma) contains fauna that is similar to that of Hadar where the proportion of bovids compared to suids and equids is not very high and also where suids are more common than equids. In contrast, the Seraitu lake beds (~2.8 to 2.6 Ma) and Uraitele zone (~2.5 to 2.4 Ma) contain more bovids while equids overtake suids. This suggest that the older fauna in Gafura might have followed the migration of the Hadar Lake Basin northeast around 3 Ma, resulting in faunal similarities with Hadar. Relative faunal abundance however seems to have been altered in the younger horizons of the Seraitu lake beds and Uraitele (after ~2.9 Ma), leading to a faunal assemblage indicative of more open conditions. This is supported by the overall abundance of bovids, particularly alcelaphins, and equids probably indicating an in situ faunal turnover.

We used all MLP specimens identifiable to genus to compute the Sørenson (also known as Dice) faunal dissimilarity index for each pairwise comparison among the faunal zones. The results indicate that Seraitu and Uraitele are more compositionally similar to one another (Sørenson = 0.31) at the genus level than either of these zones is to Gafura (Gafura-Seraitu Sørenson = 0.4, Gafura-Uraitele Sørenson = 0.44). See Table 1 for faunal abundances in the three MLP zones. We further conducted a correspondence analysis on taxon abundances in order to compare the MLP faunal zones with assemblages from the Hadar Formation at Hadar and Dikika (Fig. 5). We restricted our analysis to seven bovid tribes (Aepycerotini, Alcelaphini, Antilopini, Bovini, Hippotragini, Reduncini, Tragelaphini), the suid genera *Notochoerus* and *Kolpochoerus*, and all Equidae identified only to family. These relatively broad taxonomic categories were chosen to reduce the influence of inter-observer variation in taxonomic identifications. The correspondence analysis (Fig. 5) demonstrates that the Gafura assemblage is distinct from the Seraitu and Uraitele assemblages, with Gafura showing a high abundance of *Notochoerus*, and the Seraitu and Uraitele assemblages showing a high abundance of Alcelaphini and Antilopini, which are open-habitat indicator taxa.

It is possible that some of the observed taxonomic differences between the collecting areas of successive ages are due to small sample size, but some have clear biochronological significance.

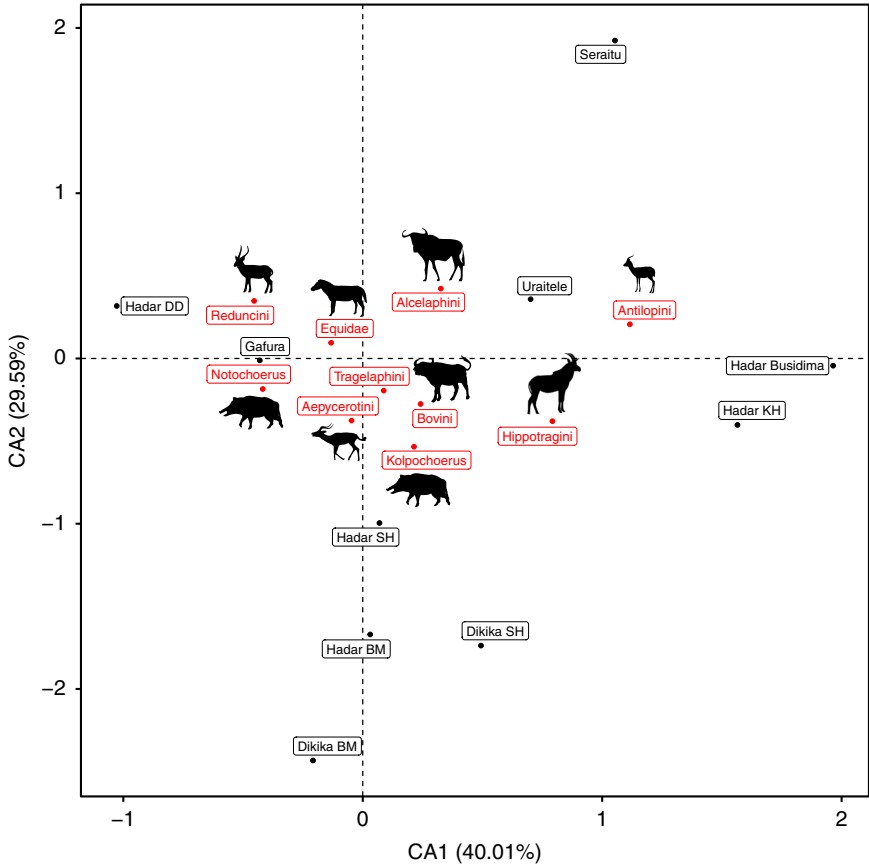

**Fig. 5 Correspondence analysis on taxon abundances comparing the MLP faunal zones with assemblages from the Hadar Formation at Hadar and Dikika.** Results demonstrate that the Gafura assemblage is distinct from the Seraitu and Uraitele assemblages, with Gafura showing a high abundance of *Notochoerus*, and the Seraitu and Uraitele assemblages showing a high abundance of Alcelaphini and Antilopini, which are open-habitat indicator taxa.

We interpret the presence of *Damaliscus* cf. *ademassui*, *Connochaetes* sp., and perhaps *Kobus sigmoidalis*, at Uraitele to suggest that the latter is younger than Gafura and Seraitu. In addition, it is not yet certain if the *Tragelaphus* from Gafura is a "typical" *T. nakuae*, and whether *Damalborea* survived into the Uraitele horizon.

In sum, the overall similarity of the Mille-Logya fauna, especially from the older Gafura sites, with that from the Hadar Formation is remarkable. Also, in spite of the apparent younger age, it contrasts with younger sites in the Middle and Lower Awash (Table 2). For instance, the bovid assemblage of Bouri Hata at ca. 2.5 Ma[52] includes a long list of bovid taxa for which there is no evidence at Mille-Logya: *Beatragus*, cf. *Numidocapra*, cf. *Rabaticeras*, *Megalotragus*, *Hippotragus*, *Oryx*, and *Tragelaphus strepsiceros*. The nearby Ledi-Geraru area contains sediments whose ages are very similar to those of Mille-Logya, but their faunal assemblage also looks different, containing *Beatragus*, *Syncerus* and *Ugandax*, but lacking *Kobus oricornus*, the most common reduncini at Mille-Logya.

In regards to paleoenvironments, tragelaphins are rare as are giraffes, while hipparions and reduncins are common. Alcelaphins are by far the most abundant bovids. The relative abundance of the otherwise rare *Camelus* is also noteworthy. On the whole, this assemblage points to an open savanna or grassland with little woody cover. This in conjunction with the presence of *Homo* at Mille-Logya may suggest that the earliest members of *Homo* were associated with more open environments than *Australopithecus* was. The in situ faunal change at Mille-Logya may be linked to environmental and climatic factors that

may have also caused *Homo* to emerge in or disperse to the region. Further field work and faunal analysis with better taxonomic resolution and use of additional proxies will help to better elucidate the paleoenvironmental and paleoecological conditions of this new site and its relevance to the understanding our origins.

Although the Afar Depression has contributed uniquely to our understanding of the biological and cultural evolution of hominins and faunal evolution more broadly over the past 6 Ma, paleontological data have been sparse in the region from the stratigraphic time interval (2.9–2.4 Ma) represented by the Mille-Logya sediments. Our work shows the unique nature of the faunal assemblage and composition at the new site. The results suggest a northeast migration of the Hadar Basin and the creation of a new depocenter at Mille-Logya with continuity of the lake deposits and shoreline sediments that preserve a fauna similar to that from Hadar. Furthermore, we have identified three different fossiliferous units in this project area suggesting an in situ faunal change. Yet, relative to the Hadar Formation fauna, which is older than 3 Ma, Mille-Logya has a large proportion of alcelaphin bovids and equids, indicating that the area likely included more open habitats after 3 Ma. The absence of early *Metridiochoerus*, *Menelikia* and *Australopithecus* from Mille-Logya suggests that these habitats may not have been suitable for these taxa. The presence of *Homo* is instead suggestive of an adaptive shift in the transition between *Australopithecus* and *Homo* to settings with overall drier and more open conditions.

Looking forward, more research at MLP will allow better documentation of the geological setting and paleontological

**Table 2 Mille-Logya composite faunal list compared with those of the Hata Member of the Bouri Formation, and the Gurumaha-Lee Adoyta area.**

| MLP | Hata Mb, Bouri Fm ca. 2.5 Ma[52] | Gurumaha-Lee Adoyta 2.8-2.5 Ma[4,11] |
|---|---|---|
| Deinotherium bozasi | Deinotherium cf. bozasi | Deinotherium bozasi |
| Elephas recki | Elephas recki shungurensis | Elephas recki |
| Orycteropus sp. | – | Orycteropus sp. |
| Ceratotherium cf. mauritanicum | – | Ceratotherium sp. |
| **Diceros sp.** | – | – |
| Hipparion s.l. sp. | Hipparion sp. | Eurygnathohippus afarense |
| – | – | Eurygnathohippus hasumense |
| **Hexaprotodon sp.** | **Hexaprotodon bruneti** | Hippopotamidae sp. |
| Kolpochoerus cf. limnetes | Kolpochoerus limnetes | Kolpochoerus cf. philippi |
| – | **Metridiochoerus andrewsi** | **Metridiochoerus andrewsi** |
| Notochoerus euilus | Notochoerus sp. | Notochoerus sp. |
| **Camelus grattardi** | – | – |
| Sivatherium maurusium | Sivatherium sp. | Sivatherium maurusium |
| Giraffa cf. camelopardalis | Giraffa sp. | – |
| Giraffa cf. gracilis | – | – |
| Tragelaphus nakuae | Tragelaphus nakuae | Tragelaphus nakuae |
| Tragelaphus sp.? | **Tragelaphus strepsiceros** | Tragelaphus gaudryi |
| – | Tragelaphus pricei | – |
| Pelorovis sp. | Ugandax coryndonae | Ugandax coryndonae |
| – | Simatherium shungurense | cf. Syncerus sp. |
| **Kobus cf. oricornus** | **Kobus kob** | **Menelikia lyrocera** |
| Kobus sigmoidalis | Kobus sigmoidalis | Kobus sigmoidalis |
| – | **Hippotragus gigas** | – |
| – | **cf. Oryx sp.** | – |
| – | **Beatragus whitei** | **Beatragus sp. nov.** |
| Connochaetes sp. | Connochaetes gentryi | Connochaetes sp. |
| Damaliscus cf. ademassui | Damaliscus ademassui | cf. Damaliscus sp. |
| Damalborea sp. | **cf. Rabaticeras arambourgi** | cf. Damalborea sp. |
| Alcelaphini indet. | **cf. Numidocapra crassicornis** | **Parmularius aff. pachyceras** |
| – | **Megalotragus kattwinkeli** | – |
| – | **Parmularius rugosus** | – |
| Aepyceros sp. | Aepyceros | Aepyceros sp. |
| **Gazella harmonae** | **Gazella janenschi** | cf. Gazella sp. |
| – | Antidorcas sp. | Parantidorcas latifrons |
| – | – | cf. Antilope sp. |
| cf. Dinofelis sp. | – | cf. Dinofelis sp. |
| cf. Felis sp. | Homotherium sp. | Homotherium sp. |
| cf. Crocuta sp. | – | Crocuta dietrichi |
| – | Genetta sp. | – |
| – | Aonyx aff. capensis | – |
| Homo indet. | Australopithecus garhi | Homo sp. |
| Theropithecus sp. | Theropithecus sp. | Theropithecus darti |

Bold taxa underline the most significant differences between MLP and any of the other two areas.

content of the poorly known post-Hadar-Dikika periods to better articulate the environmental setting of human evolution. For the first time, we now have a better understanding of the cause for local cessation of sedimentation in the Hadar-Dikika areas after approximately 2.9 Ma. The new data set will serve as the basis for within-site faunal comparison and for testing competing spatial and temporal faunal and environmental change hypotheses. The presence of hominin remains, including *Homo* associated with a diverse fauna presents a unique opportunity to address key questions that pertain to our genus and factors that led to its emergence and subsequent biological and cultural evolution.

## Methods

**Stratigraphy and sedimentology.** Stratigraphic sections were measured with Jacob's staff and Brunton compass, and mapping done with hand-held GPS- and GIS-enabled devices. Samples were prepared for diatom analysis following Battarbee et al.[53], as described in detail in Supplementary Note 1. In short, samples were placed in 30% $H_2O_2$ to remove organic matter. Samples were rinsed with deionized water and on microscope cover slips. Cover slips were mounted to slides in Naphrax, a highly refractive media and analyzed under a Leica DM2500 light microscope at ×1000 magnification.

**Basalt and Tephra Geochemistry.** Twenty-seven samples from Mile-Logya basalt samples were selected for major and trace element concentration determinations using a PANalytical 2404 X-ray fluorescence (XRF) vacuum spectrometer at Franklin and Marshall College, Lancaster, PA, USA following the techniques outlined in Mertzman[54–56], as described in detail in Supplementary Note 1. In short, the analytical technique includes the determination of ferrous iron (FeO) by standard titration methods and total volatile content (LOI). We analyzed the chemical composition of volcanic glass from 25 tephra samples from the Mille-Logya region using an electron microprobe (EMP). Major-element abundances were analyzed by wavelength-dispersive spectrometry on a JEOL 8900 Superprobe housed at the Smithsonian institution's Department of Mineral Sciences (2014 samples) or a JEOL 8900 Superprobe housed at the Carnegie Institution for Science's Geophysical Laboratory (2015 samples). For each tephra sample, 9–11 shards were analyzed. Both instruments were run with 12 kV, a 10 nA beam current, and a 10 μm spot size, conditions ideal for reducing alkali loss while obtaining reliable counts for elements such as Fe. We compared analyses of tephra from Mille-Logya with published analyses of tephras from throughout eastern Africa. We use the Borchardt coefficient[52] to identify potential correlates based on glass chemistry. For chemically-similar tephras (BC ≥ 0.85), we consider the degree of similarity, stratigraphic position and radiometric ages in evaluating tephra correlations.

**$^{40}Ar/^{39}Ar$ dating.** Four tuffs were dated by the $^{40}Ar/^{39}Ar$ method, selected for the presence of K-feldspar phenocrysts. The Hinti Mageta Tuff-2 (Sample MLP14/HM-2) is a 2–3 mm thin, medium grained, crystal-lithic tuff that occurs within a several meter thick diatomite sequence, the Iki-elu Diatomite. Sample MLP14/SR-6 is one of several thin tephras within the Seraitu Lake section, and is a massive ~20 cm thick tuff with tabular feldspar grains up to 1 mm within an altered matrix. Sample MLP14/GOY-2 is a ~10 cm thick pumiceous tuff with irregular upper and lower contacts and phenocrysts up to 1 mm. The Uraitele Tuff (sample MLP14/URT-1) is a 50 cm thick fine- to coarse-grained vitric tuff with bubble wall shards and fine pumice.

Mineral preparation was performed using standard techniques, involving gentle crushing, washing in distilled $H_2O$ and 5% HF, heavy liquid separations to isolate K-feldspar, and hand-picking to minimize the presence of inclusions and any visible imperfections. Dated K-feldspars were in the 200–1000 micron size range. K-feldspar concentrates were irradiated in the Cd-lined CLICIT position of the Oregon State University TRIGA reactor for six hours. Sanidine phenocrysts from the Alder Creek Rhyolite of California (orbitally referenced age = 1.1848 ± 0.0006 Ma[53,56]; were used as the neutron fluence monitor. Reactor-induced isotopic production ratios for these irradiations were: $(^{36}Ar/^{37}Ar)_{Ca} = 2.65 ± 0.02 × 10^{-4}$, $(^{38}Ar/^{37}Ar)_{Ca} = 1.96 ± 0.08 × 10^{-5}$, $(^{39}Ar/^{37}Ar)_{Ca} = 6.95 ± 0.09 × 10^{-4}$, $(^{37}Ar/^{39}Ar)$ K = 2.24 ± 0.16 × 10^{-4}$, $(^{38}Ar/^{39}Ar)$ K = 1.220 ± 0.003 × 10^{-2}$, $(^{40}Ar/^{39}Ar)$ K = 2.5 ± 0.9 × 10^{-4}$. Atmospheric $^{40}Ar/^{36}Ar = 298.56 ± 0.31$ and decay constants follow[57].

Following a period of several weeks of radiological 'cooling' after irradiation, the feldspars were analyzed individually by the $^{40}Ar/^{39}Ar$ technique using single-crystal incremental heating (SCIH). In the SCIH method, individual phenocrysts are incrementally heated in 4–7 steps (depending on grain size and gas yield) at progressively increasing power to fusion, to better examine the argon release patterns, drive off surficial argon in early steps, and maintain fairly consistent gas yields for better reproducibility. These detailed analyses were conducted on a Nu Instruments *Noblesse* noble-gas mass spectrometer, featuring a high-efficiency ionization source and simultaneous multi-isotope measurement using all ion-counting electron multiplier detection systems. A total of 300 SCIH steps on 91 phenocrysts from the four samples were analyzed (Table 1). 48 of these phenocrysts were rejected as candidates for complete step-heating analysis after one low-power steps, due to an obviously old xenocrystic ages or high Ca/K content, whereas the remainder (43 grains) were carried to completion. All argon measurements were performed at the Berkeley Geochronology Center. Additional details of the $^{40}Ar/^{39}Ar$ dating method as applied herein are provided in ref. [58] and are described in greater detail in Supplementary Note 1.

**Magnetostratigraphy.** Paleomagnetic samples from the Gafura and Seraitu zones were collected in 2015. Sections were trenched for measurement, description, and paleomagnetic sampling. Paleomagnetic samples were drilled using a battery-powered 2.5 cm-diameter diamond-coated bit cooled with air which was applied using a handpump. Orientation of the samples were measured using a geological compass and inclinometer - no dip correction was made for the bedding as the dip

was below 5 degrees. After cutting the samples to standard size the measurement of the natural remanent magnetization (NRM) of the specimens and the progressive demagnetization was carried out in the laboratory of Paleomagnetism and Rock Magnetism at the University of Oxford (England). A pilot set of specimens were subjected to stepwise alternating-field (AF) demagnetization at applied peak fields of 0, 5, 10, 15, 20, 25, 30, 35, 40, 50, 60, 70, and 80 mT. Thermal demagnetization was done using the following temperature steps: 20, 100, 150, 200, 250, 300, 350, 400, 450, 500, 550, 575, and 600 °C. Most measurements and demagnetization steps were performed using a 2G Enterprises DC-SQUID cryogenic magnetometer with an in-line, triaxial, alternating field (AF) demagnetizer in a shielded room at the University of Oxford (United Kingdom). One batch of samples were thermally demagnetized following temperature steps 20, 100, 150, 200, 250, 300, 350, 400, 450, 500, 550, and 600 °C at Fort Hoofddijk Paleomagnetic Laboratory of the Utrecht University (The Netherlands) on a 2G Enterprises DC SQUID cryogenic magnetometer. Thermal demagnetization was performed on an ASC thermal demagnetizer (residual field <20 nT). Natural remanent magnetization (NRM) intensities were typically several orders of magnitude higher than the instrument sensitivity (~$3 \times 10^{-12}$ Am$^2$). The results of the demagnetization were interpreted to identify the Characteristic Remanent Magnetization (ChRM) directions using Paleomagnetism.org, an online open source tool for paleomagnetic data analysis[59]. ChRM directions were calculated with a minimum of four consecutive steps (See Supplementary Table 5). Paleomagnetism.org uses a set of techniques to statistically interpret the results[60–64].

For rock magnetic purposes Isothermal Remanent Magnetization (IRM) acquisition curve up to 1T on five samples on a vibrating sample magnetometer (VSM, Micro- Mag Model 3900; Princeton Measurements).

**Reporting summary**. Further information on research design is available in the Nature Research Reporting Summary linked to this article.

## Data availability
Data are available in Supplementary information. The source data underlying Fig. 4 and Supplementary Fig. 11 are provided as a Source Data file.

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

## Acknowledgements

We thank the Authority for Research and Conservation of Cultural Heritage and the Afar Regional State for permission to conduct field work in the Mille-Logya area. We also express our gratitude to the MLP field crew and the people of the Mille and Logya towns and environs for permission and logistical support. Funding to conduct field work was provided by Margaret and Will Hearst. J.G.W. was supported by an NSF IR/D program. M.J.S. was funded by the Netherlands Organisation for Scientific Research grants NWO-ALW 823.01.003 and is currently funded by Juan de la Cierva-Incorporación grant from Spanish Research Agency, MINCIU (reference IJCI-2017-34126). Part of M.J.S. fieldwork was funded by SNMAP.nl. We are grateful Fred Spoor for data on upper molars, to A. Louchart and K. Stewart for identifying the birds and fishes, respectively, and to T. Lehmann for confirming the identification of *Orycteropus*, and to Kelsi Hurdle and Shariwa Oke for help with formatting the references.

## Author contributions

Z.A., J.W., D.G., D.R., W.A.B., S.P.M., A.D., M.A., and M.S. conducted fieldwork. Z.A., D.G., D. Reed., W.A.B., and RB described and analyzed fauna and hominins, J.W. studied geology and stratigraphy with contribution from AD (radiometric dating); M.A. (basalt geochemistry); M.S. (magnetostratigraphy); D. Roman (tephra geochemistry) and J.M. (diatoms). All authors contributed data and content used in the writing of the paper.

## Competing interests

The authors declare no competing interests.
