## [Peer Review File · Nature Communications]

Reviewers' Comments:

Reviewer #1:

Remarks to the Author:

NCOMMS 19-14456

This is a clear and useful description of the recent discoveries from Mille-Logya. It is nice to see a new area opened up and producing fossils. Overall, I have no strong complaints with the manuscript; overall it follows established protocols for describing the new material and the assertions seem reasonably well-supported. I am encouraged to see the work being done there - in general, it seems to have potential to produce some very interesting and informative fossils. I do have a few notes, however. Notes:

1. Page 9: Some clearer explanation of the taxonomy of the hominin tooth might be helpful. The authors say "Based on size and average enamel thickness (Fig. 3 e & f) in addition to diagnostic occlusal features, we attribute it to *Homo* sp.". Fair enough for the occlusal features and enamel thickness, but the size is not really helpful, according to my read of the graph, since the new tooth is in with *A. anamensis* and *A. afarensis*, which is, in turn between *H. habilis* and *H. erectus*. Perhaps cusp proportions might be more helpful.

Also on that graph, using only the species means gives us no sense of variation. Since the new M2 falls in with the *A. afarensis* and *A. anamensis*, just using the means might or might not be telling us the whole picture. Does the new molar not overlap at all with early *Homo* in terms of size?

2. The previous sentence reads: "An asymmetric and rhomboidal occlusal outline of the upper second molar has been reported to characterize *Homo erectus* and *H. habilis*^{35,35} but is rare in *A. afarensis* Kimbel et al.⁴²"

I would argue that the M2 rhomboidal structure might not be so diagnostic - it is present in AL 417 and AL 444. If it is truly different from *A. afarensis*, some quantitative expression of that fact might clarify.

Just a note: citation 42 is not Kimbel et al.

3. P. 9-10: Were the two ulnae from the same stratigraphic horizon? What are the ages?

4. P. 13-15 and Supplement: The faunal list is helpful, and the paleoecological interpretations seem reasonable, but there was no attempt to quantify the habitat in terms of open-closed or wet-dry. Since the authors draw specific environmental conclusions, I would have liked to have seen some kind of multivariate analysis of the faunal data to provide us with a sense of where the Mille-Logya fossils fall relative to other known ecological communities (fossil and extant). Perhaps this is in an accompanying paper?

Reviewer #2:

Remarks to the Author:

This manuscript provides a preliminary overview of the geology, fauna, and paleoenvironment of a new hominin-bearing site in Ethiopia, Mille-Logya. The site samples a little-known time period in the human fossil record and has the potential to contribute significantly to our understanding of the origin and evolution of *Homo*. However, the conclusions that the authors reached are not substantiated. This is mostly because of the lack of detail or analysis provided for the fauna. This is in direct contrast to the geology and age of the site, which is described and analyzed in great detail. To support their

conclusion that Homo appeared with drier/open environments and represents an adaptive shift between Australopithecus and Homo, the authors needed to show that the fauna from the older sediments are different from those of younger sediments, that there was indeed in-situ faunal change with fauna in Mille-Logya, and that the paleoenvironment was different between the older sediments and younger sediments. To do this, the authors really needed to:

1. Provide the data and results behind the relative abundances of the fauna. No numbers are ever provided – what is the number of specimens? What is the relative abundance? The authors just tell us, for example, that Gafura is similar to Hadar with higher proportions of bovids to suids and equids.
2. The authors talk about the degree of faunal similarity, but it is a qualitative rather than quantitative assessment. They really need to do some sort of faunal similarity analysis that provides us with a quantitative measure of (dis)similarity, such as Dice.
3. The authors only provide a composite faunal list even though they argue for in-situ faunal change. They need to provide faunal lists by localities and/or stratigraphic units.
4. There needs to be a more detailed comparison between the fauna and paleoenvironment between Mille-Logya and contemporaneous sites.
4. Relative abundances and faunal similarity can be quite biased unless we understand the taphonomic context of the site. As far as I can tell, no taphonomic analyses have been conducted.

Reviewers' comments:

Reviewer #1 (Remarks to the Author):

NCOMMS 19-14456

This is a clear and useful description of the recent discoveries from Mille-Logya. It is nice to see a new area opened up and producing fossils. Overall, I have no strong complaints with the manuscript; overall it follows established protocols for describing the new material and the assertions seem reasonably well-supported. I am encouraged to see the work being done there - in general, it seems to have potential to produce some very interesting and informative fossils. I do have a few notes, however.

Response: We thank the reviewer for appreciating the significance of the new site and the importance of the data presented in our paper.

Notes:

1. Page 9: Some clearer explanation of the taxonomy of the hominin tooth might be helpful. The authors say "Based on size and average enamel thickness (Fig. 3 e & f) in addition to diagnostic occlusal features, we attribute it to *Homo* sp.". Fair enough for the occlusal features and enamel thickness, but the size is not really helpful, according to my read of the graph, since the new tooth is in with *A. anamensis* and *A. afarensis*, which is, in turn between *H. habilis* and *H. erectus*. Perhaps cusp proportions might be more helpful.

Response: We agree with the reviewer that tooth size in itself is not good at discriminating taxa but our identification primarily depends on shape, occlusal features and enamel thickness as outlined in the paper. We do appreciate the recommendation by the reviewer to use cusp proportion. Yet, when Grine et al (2009)* used cusp proportion to explore taxonomic affinities of mostly isolated teeth from South Africa, upper molar cusp area was not informative enough while lower molar area discriminated taxa to some degree. Given the findings by Grine et al., and the fact that we only have one upper second molar (N=1), we still depend on shape, occlusal features and enamel thickness for taxonomic assignment. We believe that cusp proportion studies should be done based on a representative sample, which can only be undertaken when we have appropriate sample size. In the meantime, in order to clarify our description and illustrate the differences better, we have presented occlusal views of selected specimens below and this will be included in the supplementary info as Supp. Fig. 12 and this is also included here.

Also on that graph, using only the species means gives us no sense of variation. Since the new M2 falls in with the *A. afarensis* and *A. anamensis*, just using the means might or might not be telling us the whole picture. Does the new molar not overlap at all with early *Homo* in terms of size?

Response: We agree and in order to provide some sense of variation, we have presented below and in the main text (Fig. 3 e & f), data on bucco-lingual and mesio-distal dimensions of 7 species (*A. anamensis*, *A. afarensis*, *A. africanus*, *P. boisei*, *P. robustus*, *H. habilis*, *H. erectus*) as well as the *Homo* specimen from Hadar (A.L.666-1) and our upper molar (MLP-1549) using box plots (see below). The results show, as pointed out by the reviewer the tremendous overlap between *A. anamensis*, *A. afarensis*, *H. habilis* and *H. erectus*. However, the results also show that not only do the values for MLP-1549 (our specimen) OVERLAP with those of early *Homo*, they are closest to the mean for *H. habilis*. The buccolingual dimension in the new molar is actually very close to that in A.L. 666-1, a specimen that is securely assigned to early *Homo*. The analysis therefore supports our attribution of MLP-1549 to early *Homo* though we still think that with N=1, a stronger argument is still made based on features mentioned above.

Manuscript modified to reflect this.

Deleted: ¶

2. The previous sentence reads: “An asymmetric and rhomboidal occlusal outline of the upper second molar has been reported to characterize *Homo erectus* and *H. habilis*^{35,35} but is rare in *A. afarensis* Kimbel et al.⁴²”

I would argue that the M2 rhomboidal structure might not be so diagnostic – it is present in AL 417 and AL 444. If it is truly different from *A. afarensis*, some quantitative expression of that fact might clarify.

Response: We agree with the reviewer that rhomboidal occlusal outline is not unique to *Homo* but is MORE COMMON in this taxon and rare in *A. afarensis* as noted in the reference by Kimbel et al. we cited (ref. 36). It is also true that the feature is present in A.L. 417 and A.L. 444 (specimens we illustrated above) but not in most *A. afarensis*, which still means it is rare though not absent in that species. This is consistent with what we presented in our paper. We appreciate the reviewer’s recommendation to undertake additional study to quantify these subtle differences, however such study would require additional data, which would involve extensive data collection beyond the scope of this study. Since our team has CT scans of the Hadar and Omo fossils, we have just started collecting both 2D and 3D data on occlusal morphology aiming at a more comprehensive publication in a more specialized journal. We strongly feel however that such study is not required to make the points made in our paper including the assignation of the new molar to *Homo*.

In regards to the specific specimens mentioned by the reviewer, A.L. 417 and A. L.444, we have illustrated them above along with MLP-1549 and A.L. 666-1 to show the subtle but diagnostic differences mentioned in the text. This is also included in the supplementary info.

Just a note: citation 42 is not Kimbel et al.

Response: Fixed

3. P. 9-10: Were the two ulnae from the same stratigraphic horizon? What are the ages?

Response: The two Ulnae are neither from the same individual nor locality. They come from the Seraitu stratigraphic level dated ~2.8 to 2.6 Ma, which is older than the horizon that yielded the molar.

Manuscript modified to reflect this.

4. P. 13-15 and Supplement: The faunal list is helpful, and the paleoecological interpretations seem reasonable, but there was no attempt to quantify the habitat in terms of open-closed or wet-dry. Since the authors draw specific environmental conclusions, I would have liked to have seen some kind of multivariate analysis of the faunal data to provide us with a sense of where the Mille-Logya fossils fall relative to other known ecological communities (fossil and extant). Perhaps this is in an accompanying paper?

Response: please see our responses to Reviewer #2 below.

Reviewer #2 (Remarks to the Author):

This manuscript provides a preliminary overview of the geology, fauna, and paleoenvironment of a new hominin-bearing site in Ethiopia, Mille-Logya. The site samples a little-known time period in the human fossil record and has the potential to contribute significantly to our understanding of the origin and evolution of Homo. However, the conclusions that the authors reached are not substantiated. This is mostly because of the lack of detail or analysis provided for the fauna. This is in direct contrast to the geology and age of the site, which is described and analyzed in great detail. To support their conclusion that Homo appeared with drier/open environments and represents an adaptive shift between Australopithecus and Homo, the authors needed to show that the fauna from the older sediments are different from those of younger sediments, that there was indeed in-situ faunal change with fauna in Mille-Logya, and that the paleoenvironment was different between the older sediments and younger sediments. To do this, the authors really needed to:

We thank the reviewer for these constructive criticisms, which have improved the quality of the paper.

1. Provide the data and results behind the relative abundances of the fauna. No numbers are ever provided – what is the number of specimens? What is the relative abundance? The authors just tell us, for example, that Gafura is similar to Hadar with higher proportions of bovids to suids and equids.

We now include the abundance values underlying the correspondence analysis, as well as the generic counts in the supplementary information

2. The authors talk about the degree of faunal similarity, but it is a qualitative rather than quantitative assessment. They really need to do some sort of faunal similarity analysis that provides us with a quantitative measure of (dis)similarity, such as Dice.

As requested by reviewer #2, we computed the Sorenson (also known as Dice) faunal dissimilarity index and report the results in the text.

3. The authors only provide a composite faunal list even though they argue for in-situ faunal change. They need to provide faunal lists by localities and/or stratigraphic units.

See response to point #1, the revised version now provides the faunal lists underlying the analyses

4. There needs to be a more detailed comparison between the fauna and paleoenvironment between Mille-Logya and contemporaneous sites.

We agree this is a worthy goal, however this is limited by the fact that the faunal abundances for some of the most relevant contemporaneous sites have not been published. Given this constraint, we include a correspondence analysis comparing the MLP fauna with that from Hadar and Dikika (Fig. 4).

5. Relative abundances and faunal similarity can be quite biased unless we understand the taphonomic context of the site. As far as I can tell, no taphonomic analyses have been conducted.

This is a good point; we have added a few words about this in the main text. First, we must mention that our collecting procedures at MLP are standardized (Ref. 34), i.e., we always collect all cranial and dental material of bovids, suids, and equids, so there is no collecting bias that would explain the main difference that we observe within MLP.

Collecting procedures may have been less strictly standardized at Hadar, but we believe that they were broadly similar to ours.

Regarding taphonomy, it is true that taphonomic analyses have not been conducted yet at MLP, but we do not believe that preservation factors might have significantly altered the relative proportions of the various groups, because our counts are mostly based upon cranial pieces, jaw fragments, and isolated teeth that are of comparable sizes in the taxa that we used for similarity indices and CA (Bovidae, Suidae, Equidae). They are thus unlikely to have had very different depositional and post-depositional histories.

*Grine, F.E, Smith, H.F, Heesy, C.P. & Smith, E.J. 2009. Phenetic Affinities of Plio-Pleistocene Homo Fossils from South Africa: Molar Cusp Proportions. In: Grine, F.E. et al. (eds.), *The First Humans: Origin and Early Evolution of the Genus Homo*, 49 *Vertebrate Paleobiology and Paleoanthropology*, © Springer Science

Reviewers' Comments:

Reviewer #1:

Remarks to the Author:

I think the revisions have improved the document, and it is ready to be published.

Reviewer #2:

Remarks to the Author:

The revised version of the manuscript is much improved. There are just a few relatively minor comments that I think the authors can deal with easily.

1. The authors state that the BL and MD dimensions of MLP-1549 overlap with those of early Homo and are closest to their mean values. However, from the figures provided, it appears that they overlap in values with every hominin except for P. robustus. The authors seem to use this as part of their rationale for identifying the molar as early Homo even though the morphology they provide further down the paragraph is actually more convincing. It is merely a shift in emphasis I think.

2. Figure 3 - I would strongly advise the authors to make e. and f. easier to read. Please write out taxonomic names instead of labeling them with letters. Larger graphs would be very useful too. These are almost impossible to read.

3. In the discussions of implications of Mille-Logya fauna and paleoenvironment, the authors gloss over or do not even mention Ledi-Geraru, which is a significant oversight as it has the oldest known Homo specimen and provides important context for the authors' hypothesis of early Homo habitat and origin.

4. In supplementary figures 10 and 11, robustus is labeled as both A. and P., please be consistent.

REVIEWERS' COMMENTS:

Reviewer #1 I think the revisions have improved the document, and it is ready to be published.

Response: Thank you

Reviewer #2 (Remarks to the Author):

The revised version of the manuscript is much improved. There are just a few relatively minor comments that I think the authors can deal with easily.

Response: Thank you

1. The authors state that the BL and MD dimensions of MLP-1549 overlap with those of early Homo and are closest to their mean values. However, from the figures provided, it appears that they overlap in values with every hominin except for *P. robustus*. The authors seem to use this as part of their rationale for identifying the molar as early Homo even though the morphology they provide further down the paragraph is actually more convincing. It is merely a shift in emphasis I think.

Response: We agree with the reviewer on the overlap, which we had made clear since our first submission. This overlap is now clearly presented in Figure 4a&b (previously Figure 3 e&f) . However, right from the beginning, our taxonomic attribution was done based on morphological features rather than metrics on one molar crown. As already indicated in our paper we agree with the reviewer that the emphasis should be on morphology and not crown dimension which is what we do in our paper.

2. Figure 3 - I would strongly advise the authors to make e. and f. easier to read. Please write out taxonomic names instead of labeling them with letters. Larger graphs would be very useful too. These are almost impossible to read.

Response: Done. Figure 3 e&f now stands on its own as Figure 4a&b

3. In the discussions of implications of Mille-Logya fauna and paleoenvironment, the authors gloss over or do not even mention Ledi-Geraru, which is a significant oversight as it has the oldest known Homo specimen and provides important context for the authors' hypothesis of early Homo habitat and origin.

Response: We do mention or discuss Ledi Geraru (LG) on pages 9, 11, 12 and 14 of the main text to the extent possible and as deemed appropriate based on publicly available data. We did reach out to the LG PIs asking whether all LG data were available for our analysis. We learned that the PIs are undertaking detailed comparative analyses on their data and we fully respect that. We however believe we have conducted the necessary analyses needed to support conclusions made in this paper.

4. In supplementary figures 10 and 11, *robustus* is labeled as both A. and P., please be consistent.

Response: Done: The one in Suppl Figure 11 is now labeled as *P. robustus*